# Naturally Inspired Molecules for Neuropathic Pain Inhibition—Effect of Mirogabalin and Cebranopadol on Mechanical and Thermal Nociceptive Threshold in Mice

**DOI:** 10.3390/molecules28237862

**Published:** 2023-11-30

**Authors:** Kinga Sałat, Paula Zaręba, Michał Awtoniuk, Robert Sałat

**Affiliations:** 1Department of Pharmacodynamics, Faculty of Pharmacy, Jagiellonian University, 9 Medyczna St., 30-688 Krakow, Poland; 2Chair of Pharmaceutical Chemistry, Faculty of Pharmacy, Jagiellonian University, 9 Medyczna St., 30-688 Krakow, Poland; paula.zareba@uj.edu.pl; 3Institute of Mechanical Engineering, Warsaw University of Life Sciences, 166 Nowoursynowska St., 02-787 Warsaw, Poland; michal_awtoniuk@sggw.edu.pl; 4Faculty of Electrical and Computer Engineering, Cracow University of Technology, 24 Warszawska St., 31-155 Krakow, Poland; robert.salat@pk.edu.pl

**Keywords:** mirogabalin, cebranopadol, mouse models of neuropathic pain, machine learning, image analysis

## Abstract

Background: Neuropathic pain is drug-resistant to available analgesics and therefore novel treatment options for this debilitating clinical condition are urgently needed. Recently, two drug candidates, namely mirogabalin and cebranopadol have become a subject of interest because of their potential utility as analgesics for chronic pain treatment. However, they have not been investigated thoroughly in some types of neuropathic pain, both in humans and experimental animals. Methods: This study used the von Frey test, the hot plate test and the two-plate thermal place preference test supported by image analysis and machine learning to assess the effect of intraperitoneal mirogabalin and subcutaneous cebranopadol on mechanical and thermal nociceptive threshold in mouse models of neuropathic pain induced by streptozotocin, paclitaxel and oxaliplatin. Results: Mirogabalin and cebranopadol effectively attenuated tactile allodynia in models of neuropathic pain induced by streptozotocin and paclitaxel. Cebranopadol was more effective than mirogabalin in this respect. Both drugs also elevated the heat nociceptive threshold in mice. In the oxaliplatin model, cebranopadol and mirogabalin reduced cold-exacerbated pain. Conclusions: Since mirogabalin and cebranopadol are effective in animal models of neuropathic pain, they seem to be promising novel therapies for various types of neuropathic pain in patients, in particular those who are resistant to available analgesics.

## 1. Introduction

Neuropathic pain is a chronic and debilitating condition that negatively affects the quality of life of patients who suffer from advanced diabetes, viral (HIV, VZV) infections or those who are exposed to selected antitumor therapies, such as vincristine, bortezomib, taxanes and platinum-based chemotherapeutic agents (cisplatin, oxaliplatin). Typically, mechanical allodynia, thermal (heat or cold) hyperalgesia and spontaneous pain episodes are observed in neuropathic patients [1]. As shown recently, the resistance of neuropathic pain to potent analgesics, namely opioid drugs and analgesic adjuvants, is an extremely frequent phenomenon observed in more than 40% of patients receiving pain-killing drugs [2], and therefore, novel agents able to attenuate pain are strongly needed to increase the effectiveness of neuropathic pain treatment.

Analgesic drug discovery is the area significantly inspired by natural products. At present, naturally derived compounds, such as morphine or salicylic acid derivatives still belong to the most widely used analgesics worldwide. Being more or less modified and optimized to gain the most beneficial pharmacological properties, they constitute a starting point to design and develop in vivo active compounds able to attenuate various pain symptoms by influencing pain pathways involved in pain transduction, transmission, modulation and perception. Also, they can be regarded as a pattern to design synthetic or semi-synthetic libraries of agents with improved biological profiles and to perform structure–activity relationship analyses.

In recent years, two novel drug candidates, namely mirogabalin and cebranopadol, have been assessed as potential analgesics to be used in some pain types in humans [3]. Mirogabalin ([(1*R*,5*S*,6*S*)-6-(aminomethyl)-3-ethylbicyclo[3.2.0]hept3-en-6-yl]acetic acid; Figure 1) is a new gabapentanoid drug and a selective α_2_δ_1_ ligand of the voltage-gated calcium channels (VGCCs) which was recently approved in Japan for the treatment of peripheral neuropathic pain [4]. Gabapentanoids (Figure 1) are considered as the first-line treatment for neuropathic pain of various origin [5,6,7]. Both, the α_2_δ_1_ and α_2_δ_2_ subunits of VGCCs play a key role in neuropathic pain and compounds binding to the α_2_δ_1_ subunit of these channels reduce the influx of calcium ions into neurons of the central nervous system (CNS) and this results in analgesia. This class of drugs is considered as a derivative of gamma-aminobutyric acid (GABA) [7] which is known to be a key inhibitory neurotransmitter in the mammalian CNS. Reduced GABA concentrations appear to be involved in the etiology of several neurological disorders, including anxiety, chronic pain and epilepsy [8].

In the course of research on new drugs aimed at increasing the bioavailability of GABA analogs that do not penetrate the blood-brain barrier, it was proposed to improve their lipophilicity by adding lipophilic groups to the GABA carbon skeleton. This strategy led to the discovery of well-known gabapentanoids, such as gabapentin (Figure 1), a derivative of GABA with a cyclohexane ring at the 3 position, pregabalin (Figure 1), a (S)-(+)-3-isobutyl-GABA, and phenibut a 3-phenyl-GABA [7]. Specifically, mirogabalin is a 3-substituted derivative of GABA with a lipophilic 3-ethylbicyclo[3.2.0]hept-3-en-6-yl moiety, thus its structure also consists of the two main moieties as presented in Figure 1. The intensively explored structure–activity relationship studies among derivatives of mirogabalin showed that the size of the lipophilic moiety is strictly limited and bicylo[3.2.0]heptane/heptene is the most preferred structure [9].

In preclinical studies, mirogabalin demonstrated sustained analgesic effects and provided pain relief with a more favorable CNS safety profile than pregabalin. In particular, it demonstrated a superior CNS safety margin compared to pregabalin in behavioral assays which assessed the motor coordination of experimental animals. Clinical studies of mirogabalin have demonstrated its efficacy and safety in patients suffering from painful diabetic neuropathy and postherpetic neuralgia. Recently, a placebo-controlled study has assessed its efficacy and safety in patients with central neuropathic pain (ClinicalTrials.gov: NCT03901352) and based on these results, mirogabalin was additionally approved for the treatment of both central and peripheral neuropathic pain [10].

Also, cebranopadol shows some pharmacological similarity to morphine (Figure 1), a naturally derived opioid drug originally isolated from *Papaver somniferum*, and one of the most powerful and effective analgesics. The main mechanism of action of morphine is binding to the μ-opioid receptor (MOR), κ-opioid receptor (KOR) and δ-opioid receptor (DOR) [3]. For morphine, several serious adverse effects have been widely reported, such as respiratory depression, nausea, vomiting and strong addictive potential [11]. In the search for solutions to avoid these unfavorable effects, a new target was discovered, and a new class of ligands with opioid-like effects has been developed. A major achievement was the identification of the nociceptin opioid peptide receptor (NOR), which in humans is encoded via the opioid receptor-like-1 (ORL-1) gene. It is a G protein-coupled receptor with homology to classical opioid receptors but it lacks the ability to bind opioid ligands [12]. NOR shares partial homology to MOR, KOR and DOR, and it is insensitive to opioid agonists, such as morphine. The studies on NOR agonists started with compounds based on the cyclohexanone moiety [11,13,14]. Further research demonstrated that compounds related to 4-phenylcyclohexylamine with the similarity to pethidine (Figure 1), a synthetic opioid pain medication of the phenylpiperidine class, show analgesic activity [13]. Finally, drug development studies led to discovery the cebranopadol (*trans*-6′-fluoro-4′,9′-dihydro-*N*,*N*-dimethyl-4-phenyl-spiro-[cyclohexane-1,1′(3′*H*)-pyrano[3,4-b]indol]-4-amine) a novel analgesic drug candidate, a NOR and opioid receptor agonist. Cebranopadol acts as a full agonist of MOR (Ki = 0.7 nM) and DOR (Ki = 18 nM) and as a partial agonist of NOR (Ki = 0.9 nM) and KOR (Ki = 2.6 nM) [15,16,17]. The similarity of the MOR and NOR binding sites allows for similar π-π stacking interactions of aromatic moieties of ligands, and hydrophobic cavities are conserved in both receptors [18]. The tertiary amine of cebranopadol, which is protonated under physiological conditions, forms an ionic interaction in the binding site of NOR. Additionally, the hydrophobic pocket in the binding site of the receptor is occupied by the phenyl moiety of cebranopadol. These two functional groups, a tertiary amine group and the phenyl group, create the basic pharmacophore [11]. The structural moieties discussed above are also responsible for morphine binding to the MOR. The morphine molecule is composed of five fused rings, in which only one ring is aromatic, the second consists of the piperidine ring with a tertiary nitrogen atom giving the molecule a basic character, and the other two are fused cyclohexene rings. The aromatic ring and the cyclohexene ring are linked by an epoxide bridge, which can be thought of as the fifth dihydrofuran ring. The key fragment in the structure of morphine is the 4-(2-aminoethyl)phenol (moiety marked in light blue, Figure 1). The protonated nitrogen atom creates ionic interactions and the phenolic group in the 3-position creates hydrogen bonds and determines the weak acidic properties of the structure, while the aromatic ring creates van der Waals interactions with hydrophobic amino acids in the active site of the receptor [19]. Naturally derived opioids (morphine), as well as synthetic ones (levorphanol, metazocine), possess a phenolic hydroxyl group. However, it is of note that the high potency of cebranopadol does not require this functional group [14].

Cebranopadol possesses high permeability into the CNS and its significant analgesic, antiallodynic and antihyperalgesic properties were demonstrated in several rodent models of acute nociceptive, inflammatory, cancer and neuropathic pain [3,17]. In contrast to classical opioids, it has better efficacy in several models of neuropathic pain than in acute pain with a limited risk to produce opioid-like adverse effects, e.g., physical dependence [3].

Considering the pharmacological and chemical profiles of mirogabalin and cebranopadol, the aim of the present research was to assess how these two promising agents affect pain threshold in neuropathic mice. To study their effect on mechanical and thermal hypersensitivity, we used three mouse models of neuropathic pain, i.e., a diabetic neuropathic pain model induced by streptozotocin (STZ), a paclitaxel-induced pain model and a oxaliplatin-induced neuropathic pain model. In mice, pain induced by STZ reflects neuropathic pain in the course of advanced human type I diabetes, while pain caused by paclitaxel or oxaliplatin mimic symptoms of neuropathic pain that occur in patients receiving these antitumor drugs. The latter clinical entity is called chemotherapy-induced peripheral neuropathy (CIPN).

## 2. Results

### 2.1. STZ Model

#### 2.1.1. Effect of Mirogabalin and Cebranopadol on the Mechanical Nociceptive Threshold (von Frey Test)

In the von Frey test carried out in diabetic neuropathic mice, repeated measures ANOVA revealed an overall effect of treatment (F[4, 90] = 52.58, *p* < 0.0001). Time effect and the drug × time interaction were also statistically significant (F[1, 90] = 38.80, *p* < 0.0001 and F[4, 90] = 14.69, *p* < 0.0001, respectively).

As shown in Figure 2A, in mice, STZ lowered pain threshold for mechanical stimulation (*p* < 0.0001). In this assay, mirogabalin at the dose of 30 mg/kg and cebranopadol at the dose of 10 mg/kg were able to reduce tactile allodynia in mice exposed to STZ (*p* < 0.0001 vs. predrug paw withdrawal threshold in the individual group and *p* < 0.0001 vs. diabetic control group).

#### 2.1.2. Effect of Mirogabalin and Cebranopadol on the Heat Nociceptive Threshold (Hot Plate Test)

In the hot plate test carried out in diabetic neuropathic mice, repeated measures ANOVA revealed an overall effect of treatment (F[4, 80] = 7.855, *p* < 0.0001). Time effect and the drug x time interaction were also statistically significant (F[1, 80] = 22.98, *p* < 0.0001 and F[4, 80] = 10.55, *p* < 0.0001, respectively).

As shown in Figure 2B, in STZ-treated mice, heat hyperalgesia was not observed but in the hot plate test, similarly to the von Frey test, mirogabalin at the dose of 30 mg/kg and cebranopadol at the dose of 10 mg/kg were able to elevate the heat nociceptive threshold of mice (*p* < 0.01 vs. predrug latency to pain reaction in the individual group and *p* < 0.001 vs. diabetic control group).

### 2.2. Paclitaxel Model

#### 2.2.1. Effect of Mirogabalin and Cebranopadol on the Mechanical Nociceptive Threshold (von Frey Test)

The effect of mirogabalin and cebranopadol on pain symptoms in neuropathic mice exposed to paclitaxel was evaluated at two distinct time points of testing, i.e., on experimental days 1 and 7. On these days, paclitaxel was administered to mice and then, pain tests were carried out. In the von Frey test, repeated measures ANOVA revealed an overall effect of treatment (F[4, 156] = 51.43, *p* < 0.0001). Time effect and the drug x time interaction were also statistically significant (F[3, 156] = 16.40, *p* < 0.0001 and F[12, 156] = 4.516, *p* < 0.0001, respectively).

As shown in Figure 3A, on both test days, paclitaxel significantly (*p* < 0.001) lowered the mechanical nociceptive threshold in mice. In the von Frey test carried out in paclitaxel-treated mice, mirogabalin at the dose 30 mg/kg was significantly (*p* < 0.01) effective as an antiallodynic agent. Also, cebranopadol (10 mg/kg) significantly (*p* < 0.0001) reduced tactile allodynia in paclitaxel-treated mice. Both these drugs were effective on days 1 and 7 of testing.

#### 2.2.2. Effect of Mirogabalin and Cebranopadol on the Heat Nociceptive Threshold (Hot Plate Test)

The effect of mirogabalin and cebranopadol on paclitaxel-induced pain symptoms in response to thermal (heat) stimulation was assessed in the hot plate test. Repeated measures ANOVA revealed an overall effect of treatment (F[4, 158] = 13.26, *p* < 0.0001). Time effect and the drug x time interaction were also statistically significant (F[3, 158] = 13.19, *p* < 0.0001 and F[12, 158] = 3.4, *p* < 0.001, respectively).

As shown in Figure 3B, on both test days, paclitaxel did not affect significantly the heat nociceptive threshold in mice. On day 7, in paclitaxel-treated mice, mirogabalin at the dose of 30 mg/kg significantly (*p* < 0.05) prolonged latency to pain reaction in response to thermal stimulation. Also, cebranopadol (10 mg/kg) was significantly (*p* < 0.001) effective but its activity was noted only on day 1 of testing.

### 2.3. Oxaliplatin Model

#### 2.3.1. Effect of Mirogabalin on the Thermal Place Preference of Mice (Two-Plate Thermal Place Preference Test)

As shown in Figure 4A–F, when exposed to two distinct temperatures from the range between 0 and 30 °C, the mice not treated with mirogabalin showed a statistically significant (*p* < 0.05) preference (measured as longer time spent on a plate) for a zone with a higher temperature of the two that they could choose. Above this thermal range, i.e., between 30 and 35 °C (Figure 4G), as well as between 40 and 45 °C (Figure 4I), a slight (statistically not significant) preference towards a plate with a lower temperature was observed.

At some thermal ranges, the administration of mirogabalin at the dose active in the previously described neuropathic pain models (i.e., 30 mg/kg) reversed the thermal preference of mice as compared to that observed before its administration. This reversal of thermal preference (i.e., the observed postdrug preference towards a plate set at a lower temperature) was noted at the range 0 and 5 °C (*p* < 0.01; Figure 4A). When the two plates were set at 10 and 15 °C (Figure 4C) and 15 and 20 °C (Figure 4D), the preference towards a plate set at a higher temperature was observed but simultaneously, the time spent on the plate with a lower temperature set was longer after mirogabalin treatment as compared to that measured before its administration (*p* < 0.05—Figure 4C and *p* < 0.0001—Figure 4D). At thermal ranges 30–35 °C (Figure 4G) and 35–40 °C (Figure 4H), mirogabalin caused a statistically significant (*p* < 0.0001 vs. predrug measurement) preference for the plate set at a higher temperature. At the thermal range 40–45 °C (Figure 4I), mirogabalin-treated mice showed preference towards the plate with a lower temperature set (*p* < 0.0001) but the time spent on a plate set at 40 °C measured after the mirogabalin injection did not differ significantly from that measured before its administration (Figure 4I).

#### 2.3.2. Effect of Cebranopadol on the Thermal Place Preference of Mice (Two-Plate Thermal Place Preference Test)

As shown in Figure 5A–H, when exposed to two distinct temperatures from the range between 0 and 40 °C, mice, before cebranopadol administration, showed a statistically significant (*p* < 0.001) preference (i.e., a longer time spent on a plate) for a zone with a higher temperature of the two that were tested. Above this thermal range, i.e., between 40 and 45 °C (Figure 5I), a statistically significant preference for the plate set at 45 °C was observed only in oxaliplatin-treated mice (*p* < 0.0001 vs. time on a plate set at 40 °C) and not before oxaliplatin administration (Figure 5I).

At some thermal ranges, the administration of cebranopadol at the dose of 10 mg/kg partially reduced the thermal preference of mice as compared with that observed before its administration. This effect on thermal preference was noted at ranges 0 and 5 °C (*p* < 0.0001; Figure 5A), 5 and 10 °C (*p* < 0.0001; Figure 5B), 10 and 15 °C (*p* < 0.001; Figure 5C), 15 and 20 °C (*p* < 0.0001; Figure 5D), 20 and 25 °C (*p* < 0.05; Figure 5E), 25 and 30 °C (*p* < 0.01; Figure 5F). When the two plates were set at temperatures ranging between 30 and 45 °C (Figure 5G–I), the preference towards a plate set at a higher temperature was still noted (similar to as before cebranopadol administration, *p* < 0.0001).

## 3. Discussion

Neuropathic pain is a chronic pain type which is often resistant to available analgesics [20,21,22] and therefore, much effort is made to find novel methods to cope with this serious clinical issue. In this context, there is a strong need for new analgesic drug candidates able to attenuate the main symptoms of neuropathic pain, i.e., mechanical and thermal hypersensitivity resulting in tactile and thermal allodynia and hyperalgesia [23,24].

Both mirogabalin and cebranopadol seem to be potential novel treatment options for neuropathic pain. Although they have been found effective and well-tolerated in clinical trials in patients suffering from various types of chronic and acute postoperative pain [25,26,27,28,29,30,31,32,33,34,35,36,37,38], there is very limited data about their efficacy at the preclinical level, i.e., in some experimental models of pain resulting from drug-induced injuries to peripheral sensory nerves. For this reason, this in vivo research was focused on the assessment of the effects of mirogabalin and cebranopadol on the pain threshold in mice exposed to neuropathy-inducing drugs.

We used three mouse models of neuropathic pain, namely diabetic neuropathic pain induced by STZ (a model of painful peripheral neuropathy often present during the advanced stage of diabetes type 1 [39]), and two CIPN models induced via antitumor drugs: a taxane derivative (paclitaxel) and a platinum-based alkylating agent (oxaliplatin), whose main adverse effects are related to impaired thermo-and mechanosensation, both in humans and experimental animals [40,41].

In the mouse STZ-induced model of diabetic neuropathic pain, we confirmed that mirogabalin used at the dose of 30 mg/kg was able to elevate the mechanical and heat nociceptive threshold. This finding is in line with the previously reported data from diabetic patients (e.g., [26,29]) and some limited data about its preclinical efficacy in rodent models of diabetic peripheral neuropathy (e.g., [42]), as well as chronic constriction injury [43] and spinal cord injury [44] animal models. In our research, a lower dose of mirogabalin (10 mg/kg) compared to STZ-treated control was also able to attenuate tactile allodynia in diabetic neuropathic mice but this dose did not affect the heat nociceptive threshold.

Of note, in our present study, the pharmacological profile of the effectiveness of mirogabalin in the STZ model was similar to that previously shown for its analogue, namely pregabalin (also a ligand of α_2_δ subunit of VGCCs). Both agents statistically significantly influenced the mechanical and thermal sensitivity threshold in diabetic, STZ-treated mice [45]. Their activity in this neuropathic pain model seems to result from the effect on VGCCs as binding to the VGCC α_2_δ subunit underlies the usefulness of its ligands in some clinical disorders, including epilepsy, pain in the course of diabetic neuropathy, postherpetic neuralgia, fibromyalgia and generalized anxiety disorder. Thus, the modulation of the α_2_δ subunit of the VGCCs results in a reduction in the excessive neurotransmitter release that is observed in these neurological and psychiatric conditions [46].

For cebranopadol, there is also a limited number of data about its effectiveness in a diabetic painful peripheral neuropathy model. A study by Tzschentke and colleagues [47] revealed its antihyperalgesic, antiallodynic and antinociceptive properties in several rat and mouse models of neuropathic pain, which suggested that cebranopadol might be a potential treatment option for chronic pain with a neuropathic component.

First, we investigated its activity in the mouse model of diabetic neuropathic pain. The dose assessed in this in vivo study (10 mg/kg) was chosen based on our previous experiments [17]. This dose statistically significantly affected the mechanical and thermal (heat) nociceptive threshold in diabetic, STZ-treated mice. The potential mechanism underlying the observed effectiveness of cebranopadol in diabetic neuropathic pain is thought to be due to its effect on NOR and MOR [48,49].

In contrast to mirogabalin, in the present study, a higher dose of cebranopadol (i.e., 30 mg/kg) could not be assessed in view of the adverse effects previously observed for this dose (“Straub tail”, sedation, neurological impairments observed in mouse locomotor activity and the rotarod tests).

Of note, the results of this part of the present experiment allowed, for the first time, a direct comparison to be made of the analgesic efficacy between mirogabalin and cebranopadol under the same laboratory conditions. This analysis showed that in the diabetic model of neuropathic pain, cebranopadol was effective at the dose at which mirogabalin did not show activity. This was particularly demonstrated in the hot plate test carried out in STZ-treated mice. The hot plate test is regarded as a rodent model of thermally induced pain. In this assay, analgesics acting by peripheral mechanisms are generally not active [50]. The characteristic response which occurs in this test (jumping, licking of the paws) is of central origin and drugs with antinociceptive properties in the hot plate test act primarily in the spinal medulla and/or higher central nervous system levels [50]. Thus, in line with our previous experiments [17], the results obtained in the present study confirmed the role of the central opioidergic system in inducing analgesia caused by cebranopadol.

To the best of our knowledge, for the first time, we also compared the analgesic activity of mirogabalin and cebranopadol in a mouse model of paclitaxel-induced neuropathic pain. In mice and rats paclitaxel, a taxane derivative, induces CIPN with hypersensitivity to mechanical and thermal stimuli. Hence, its application to rodents is one of several well-established methods to investigate analgesic, antiallodynic and antihyperalgesic properties of drug candidates able to attenuate neuropathic pain in the course of CIPN [51,52]. The mechanisms underlying paclitaxel-induced cytotoxicity to sensory nerves are complex and they comprise, inter alia, altered opioidergic neurotransmission signaling [53,54], oxidative stress-induced mitochondrial damage, dysregulated calcium homeostasis, neuroinflammation, as well as changed neuronal excitability due to overexpressed voltage-gated ion (calcium and sodium) channels [41,55,56]. Considering this, in the paclitaxel-induced CIPN model, we focused on the assessment of antiallodynic and antihyperalgesic potential of VGCC ligand (mirogabalin) and we compared its activity to that of opioidergic agonist (cebranopadol).

The onset of sensory symptoms after a single injection of paclitaxel appears within hours after its administration but this effect might be unstable and transient (until 24 h after a single dose of paclitaxel). Therefore, in our research, we decided to use two doses of paclitaxel administered to mice on days 1 and 7 to achieve full hypersensitivity which is usually noted between 7 and 14 days after a single injection of this drug [57]. The main effect noted due to paclitaxel administration to mice was increased sensitivity to mechanical stimulation (tactile allodynia). In contrast to this, in our study, heat hyperalgesia in paclitaxel-exposed animals was not observed, which is in line with some previous reports showing that, in this pain model, heat hyperalgesia might not occur [55,58,59] as the symptoms observed strongly depend on the animal strain, as well as different methodology for CIPN induction used in laboratories [59].

In paclitaxel-treated mice, tactile allodynia was attenuated by mirogabalin at the dose of 30 mg/kg and by cebranopadol. This antiallodynic effect was noted at both time-points of testing. On day 1, the lower dose of mirogabalin was also slightly effective as compared to control mice injected with paclitaxel. Heat pain threshold was significantly affected by mirogabalin 30 mg/kg and by cebranopadol but for both agents this effect was noted at distinct time points (day 1—cebranopadol, and day 7—mirogabalin) suggesting that in paclitaxel-exposed mice, opioidergic mechanisms and VGCC expression might modulate the thermal nociceptive threshold differentially and at distinct stages of neuropathy. Of note, in both behavioral assays, cebranopadol was more effective as compared to mirogabalin used at the dose 3-fold higher. These findings are in line with the previously reported data [53,54] showing that opioid receptors might play a key role in reducing pain related to CIPN caused by paclitaxel.

An evident impact of mirogabalin and cebranopadol on the mechanical nociceptive threshold and not entirely clear and unequivocal results obtained for these drugs in behavioral tests assessing their impact on the thermal (heat) nociceptive threshold, turned out attention to a wide-range measurement of their effect on animals’ thermal preference. The effect of mirogabalin and cebranopadol on animals’ thermal preference has not been previously described. For this purpose, we used an additional CIPN model, i.e., the oxaliplatin-induced model, and we focused only on measuring the thermal nociceptive threshold in mice exposed to oxaliplatin. The effect of this agent on mechanical allodynia was reported previously (e.g., [60,61,62]).

Oxaliplatin is a third-generation platinum derivative which is used in combination with other cytotoxic drugs as the first-line treatment and as adjuvant therapies for colorectal cancer. Similar to other platinum-based compounds (e.g., cisplatin, carboplatin), oxaliplatin inhibits the proliferation of tumor cells by forming deoxyribonucleic acid (DNA)-platinum complexes, and this destroys cancer cells. In vivo oxaliplatin is rapidly and non-enzymatically metabolized into several derivatives, including oxalate, monochloro, dichloro- and diaquo-diaminocyclohexane platinum metabolites, and oxalate seems to be responsible for the cold-induced hypersensitivity frequently observed in patients undergoing oxaliplatin therapy. Oxalate-induced cold-exacerbated pain is thought to be a consequence of extracellular calcium ion chelation, which in turn increases sodium conductance, stimulates neuronal depolarization and hyperexcitability of sensory neurons [63].

In this part of the present research, we assessed the wide-range (0–45 °C) thermal preference of mice before oxaliplatin, mirogabalin and cebranopadol administration and then after their injection. A loss of preference for a zone with a higher temperature, or a reversal of preference towards a zone with a lower temperature of the plate after mirogabalin or cebranopadol administration was regarded as a measure of their potential analgesic effect. When mice were exposed to temperatures 0 and 5 °C, the administration of mirogabalin reversed the previously observed preference towards a plate set at the higher temperature tested, and after mirogabalin, the time spent on the plate set at 0 °C was significantly longer as compared to that before mirogabalin administration. Of note, this effect caused by mirogabalin was not observed when the plates were set at 5 and 10 °C. At these temperatures, the preference of mice towards a warmer plate (10 °C) was maintained. This finding is potentially interesting and it confirms that in mice (in contrast to rats), to obtain reliable results in pain tests, the assessment of cold pain threshold should be carried out at temperatures below 5 °C [64].

Recently, the assessment of the optimal temperature for housing laboratory animals has become a subject of growing interest, and animal welfare is one of the main reasons for the increased research focused on establishing the thermal preference of mice. Using a variety of methods to study behavioral thermoregulatory responses, it has been demonstrated that non-neuropathic mice generally prefer temperatures maintained at 22–30 °C, while those of 34 °C and higher might be recognized as heat stress [65]. A rat study by Allchorne and colleagues [66] showed that in naïve rodents, the threshold for eliciting cold pain behavior is 5 °C, while temperatures 10–25 °C are innocuous for non-neuropathic subjects but they might be potentially harmful for neuropathic animals with symptoms of allodynia. In this previous study, for neuropathic animals, the sensitivity threshold for cold allodynia was established at 15 °C. At temperatures 15 and 20 °C, where cold allodynia is noted [66], mirogabalin reversed the thermal preference of oxaliplatin-treated mice. Taken together, in our study we demonstrated that mirogabalin was able to reduce cold hyperalgesia and cold allodynia in oxaliplatin-exposed mice.

At temperatures within the thermal preferendum (20–30 °C), mirogabalin had no effect on the thermal preference of oxaliplatin-treated mice. At higher thermal ranges (30–40 °C), mirogabalin reversed thermal preference of mice treated with oxaliplatin, which might be considered as a measure of its analgesic activity. Of note, an undisturbed thermal preference of oxaliplatin-treated mice injected with mirogabalin was noted at the highest thermal range tested (40 and 45 °C). This avoidance of high temperatures, which are potentially harmful to the skin, may indicate an undisturbed perception of painful thermal stimuli (i.e., heat) in mirogabalin-exposed mice. This, in turn, should be regarded as an important feature of the drug tested.

In the experiment that used two plates set at 0 and 5 °C, in contrast to the observed before cebranopadol administration preference of mice for a warmer plate, the mice injected with cebranopadol did not show preference towards a plate set at a higher temperature. The time spent on the plate set at 0 °C was significantly longer as compared to that before cebranopadol administration. This indicated the potential analgesic properties of cebranopadol in oxaliplatin-treated mice and showed its ability to elevate the cold pain threshold in these mice. A similar effect of cebranopadol on pain threshold was also demonstrated in our earlier study that used the mouse cold plate test [17].

In contrast to mirogabalin, cebranopadol also induced a loss of thermal preference of oxaliplatin-treated mice when the plates were set at 5 and 10 °C, so its analgesic activity seems to have a broader thermal range than that observed for mirogabalin. At temperatures below (10–20 °C) and within the thermal preferendum (20–30 °C), cebranopadol was only slightly effective in reversing animals’ thermal preference and longer time spent on a warmer plate of the two used was still noted. At temperatures 30–45 °C, the thermal preference of mice before and after cebranopadol administration did not differ significantly, indicating that, at this thermal range, cebranopadol did not influence the heat pain threshold of oxaliplatin-treated mice.

## 4. Materials and Methods

### 4.1. Animals and Housing Conditions

Adult male Albino Swiss (CD-1) mice were used in the behavioral experiments. The animals were kept in groups of 10 mice in cages at a room temperature of 22 ± 2 °C, under light/dark (12:12) cycle and had free access to food (Rodentia Basic, Anima-Vivari, Warsaw, Poland) and water before experiments. The ambient temperature of the room and humidity were kept consistent throughout all the tests. For the experiments, the animals were randomly selected. Each experimental group consisted of 7–10 animals/dose. The experiments were performed between 8 AM and 2 PM. In this in vivo study, anesthesia was not used at any stage in the experiment because both general and local anesthetics could significantly influence the results of behavioral tests assessing the pain threshold in animals, thus giving false positive results in pain tests. After the assays, the animals were euthanized by cervical dislocation which is a rapid method and does not require using anesthetic drugs. Experimental procedures for in vivo tests were approved by the Local Ethics Committee in Krakow and the treatment of animals was in full accordance with the ethical standards laid down in the respective Polish and EU regulations (Directive 2010/63/EU).

### 4.2. Chemicals

STZ and paclitaxel were supplied by Sigma Aldrich (Poznań, Poland) and oxaliplatin was purchased from Activate Scientific GmbH (Chiemsee, Germany). For behavioral experiments, mirogabalin (MedChemExpress, Sollentuna, Sweden) and cebranopadol (MedChemExpress, Sweden) were suspended in a 1% Tween 80 solution (Polskie Odczynniki Chemiczne, Gliwice, Poland). In pain tests, doses 10 mg/kg and 30 mg/kg of mirogabalin administered intraperitoneally were tested. Cebranopadol at the dose of 10 mg/kg was injected subcutaneously. The doses of mirogabalin and cebranopadol were selected based on the previous studies [17] and the available literature data [67]. Control animals were given an appropriate amount of vehicle (1% Tween 80).

### 4.3. Behavioral Testing Protocol

#### 4.3.1. Induction of Neuropathy—STZ Model (Diabetic Neuropathy Model)

To induce type I diabetes, the mice were intraperitoneally injected with STZ (a single dose of 200 mg/kg) dissolved in 0.1 N citrate buffer (Polskie Odczynniki Chemiczne, Poland). Age-matched control mice received an equal volume of citrate buffer [68]. The blood glucose level was measured 20 days after STZ injection. For this purpose, a blood glucose monitoring system (AccuChek Active, Roche, Boulogne-Billancourt, France) was used. Blood samples (5 µL/mouse) were obtained from the tail vein of the mice. The animals were considered as diabetic when their blood glucose concentration exceeded 300 mg/dL [68]. Further pain tests were carried out 24 h after the selection of diabetic animals (3 weeks after STZ injection) [45].

#### 4.3.2. Induction of Neuropathy—Paclitaxel Model (CIPN Model)

In this neuropathic pain model, to induce peripheral neuropathy, paclitaxel was prepared in a 0.9% saline solution (Polfa Kutno, Kutno, Poland) and was administered as an intraperitoneal dose of 6 mg/kg at two time points, i.e., on days 1 and 7, on each test day 3 h before predrug measurements of the pain threshold [61,69].

#### 4.3.3. Induction of Neuropathy—Oxaliplatin Model (CIPN Model)

In this neuropathic pain model, to induce peripheral neuropathy, oxaliplatin was prepared in a 5% glucose solution (Polfa Kutno, Poland) and was administered as a single intraperitoneal dose of 10 mg/kg 3 h before predrug measurements of pain threshold [70,71,72].

#### 4.3.4. Assessment of Mechanical Nociceptive Threshold (von Frey Test)

To assess the effect of mirogabalin and cebranopadol on tactile allodynia in neuropathic (STZ-treated or paclitaxel-treated) mice, the von Frey test was used. Tactile allodynia was assessed using the electronic von Frey unit (Bioseb, Vitrolles, France). This apparatus is supplied with a single flexible filament applying increasing force (from 0 to 10 g) against the plantar surface of the hind paw of a mouse. The nocifensive paw withdrawal response automatically turns off the stimulus and the mechanical pressure that evoked the response is recorded.

On the day of the experiment (on day 21 after STZ injection, and on days of paclitaxel injection), the mice were placed individually in test compartments with a wire mesh bottom and were first allowed to habituate for 1 h. After the habituation period, in order to obtain baseline (predrug) values of pain sensitivity, each mouse was tested 3 times alternately in each hind paw, allowing at least 30 s between each measurement. Then, the mice were pretreated with mirogabalin, cebranopadol or vehicle, and later, at the previously established time points of testing: 60 min after mirogabalin or 90 min after cebranopadol, the animals were tested again to obtain postdrug values of pain sensitivity (postdrug mechanical nociceptive threshold) [45].

#### 4.3.5. Assessment of Heat Nociceptive Threshold (Hot Plate Test)

This assay was carried out in STZ-treated and paclitaxel-treated mice immediately after the von Frey test. The hot plate apparatus has an electrically heated surface and is supplied with a temperature-controller that maintains the temperature at 55–56 °C. The time until the animal licks its hind paws or jumps is recorded by means of a stop-watch. In this assay, the cut-off time is established (60 s) to avoid paw tissue damage, and mice not responding within 60 s are removed from the apparatus and are assigned a score of 60 s.

In the hot plate test, the animals were placed on the hot plate apparatus (Hot/cold Plate, Bioseb, France) and baseline (predrug) latencies to pain reaction were collected for each mouse. Then, postdrug latencies to pain reaction were recorded at the same time point as that in the von Frey test [45].

#### 4.3.6. Assessment of Thermal Nociceptive Threshold (Two-Plate Thermal Place Preference Test)

Thermal preference of mice was assessed according to a method previously described [70]. One day before the proper behavioral assay, mice were habituated to the analgesiameter and test conditions (room temperature: 22 ± 2 °C on both plates; habituation period: 5 min/mouse). On the test day, the thermal preference of mice was assessed at 3 distinct time points, i.e., before oxaliplatin administration (referred to as “before oxaliplatin”), 3 h after oxaliplatin (“predrug”) and then 1 h after the administration of mirogabalin or 90 min after cebranopadol administration (“postdrug”). For this purpose, each mouse was placed on a starting plate (i.e., in each experimental session, a starting plate was a plate with a lower temperature set). Then, the mouse was allowed to explore both plates of the device for 5 min. Time spent by the mouse in each thermal zone, i.e., on each plate of the device was an indicator of its thermal preference. It was recorded using a video camera (GoPro Hero7 Black, Monterey, CA, USA) and then analyzed using support vector classification (SVC) and deep learning methods [73]. The two-plate thermal place preference test was carried out in nine separate sessions, i.e., in thermal zones of the two plates ranging between 0 and 45 °C, adjustable every 5 °C, i.e., 0–5, 5–10, 10–15, 15–20, 20–25, 25–30, 30–35, 35–40 and 40–45 °C (sessions 1, 2, 3, …, etc.).

#### 4.3.7. Data Analysis

The analysis of the in vivo results was provided by GraphPad Prism Software (v.9.0, San Diego, CA, USA). The results were statistically evaluated using repeated measures analysis of variance (ANOVA), followed by Dunnett’s, Tukey’s or Sidak’s post hoc comparisons. In each case, *p* < 0.05 was considered significant.

In the thermal place preference test, the analysis of time spent in the individual thermal zone was carried out according to a method previously described [69]. In short, we used supervised machine learning [74], which helped in the image analysis, as well as in subsequent statistical analyses. The image analysis used classical analytical methods supported by advanced methods based on machine learning (e.g., deep learning). The image analysis was based on the concept of a background subtraction algorithm [75]. Step by step, the program determined whether the mouse was on the left or right plate of the device. The algorithm loaded two sequential frames into memory and proceeded with image processing: resizing, grayscale conversion and median filtering. In the next step, the image was subtracted and the result was binarized and median filtered to remove noise. In the last step, the algorithm counted white pixels (i.e., with a value of one) separately in the left and right parts of the image. The higher value denoted the area into which the presence of a mouse was classified. The algorithm was continued until the last frame of the recording.

## 5. Conclusions and Future Outlook

To sum up, our present research was the first one that was focused on the investigation of effects of mirogabalin and cebranopadol on pain responses of paclitaxel and oxaliplatin-exposed mice. We focused on studying animals’ behavioral responses and we did not assess the impact of mirogabalin or cebranopadol on various molecular targets implicated in chronic pain development: receptors, ion channels and cells implicated in pain processing (e.g., VGCCs, transporters, chemokines and microglia), as such analyses are relatively well-described in the literature (e.g., [3,7,43,76,77,78]).

The results obtained in the present in vivo research proved that mirogabalin and cebranopadol should be considered as promising novel therapies for various types of neuropathic pain. In particular, they seem to be valuable in terms of their future use in paclitaxel-treated, oxaliplatin-treated and diabetic patients who are resistant to available analgesics. Also, the results of this study may be a starting point for the drug discovery process to develop novel compounds with potential application for neuropathic pain inhibition. It should be particularly emphasized that both mirogabalin and cebranopadol possessed antiallodynic properties in the two mouse models of neuropathic pain, and in addition to this, in the oxaliplatin CIPN model, they significantly influenced thermal nociceptive threshold at temperatures at which cold hypersensitivity (both cold allodynia and cold hyperalgesia) is observed. This key finding seems to be important in terms of their potential use in clinical conditions accompanied by cold-exacerbated pain, for example in patients with oxaliplatin-induced neuropathic pain.

One should, however, note that the effect of both drugs on the cold pain threshold observed in the two-plate thermal place preference test must be also interpreted with caution, as this loss of thermal preference, or its reversal at temperatures below thermal preferendum might be potentially harmful. Nociception is a part of mechanical, thermal or chemical perception and its key role is to protect the organism against potentially harmful stimuli. Therefore, the preference for the cold plate observed in drug-treated mice might be also responsible for a diminished protection against such harmful thermal (cold) stimuli.

The results obtained in the present in vivo study might be also a starting point for the further extended investigation of the pharmacological activity of mirogabalin and cebranopadol. Neuropathic pain may have an inflammatory component [79,80] and therefore, the importance of an inflammation-related protein, namely cyclooxygenase 2 (COX2) in neuropathic pain seems to be increasing in recent years [79,81,82]. Also, considering that mirogabalin was shown to potentiate the analgesic effect of nonsteroidal anti-inflammatory drugs—inhibitors of this enzyme [83], quantitative studies assessing the effect of mirogabalin and cebranopadol on COX2 levels might be interesting. Such immunomodulatory and anti-inflammatory effects were shown in our previous study for another gabapentanoid, i.e., pregabalin which significantly lowered the expression of COX2, PGES, and NF-κB p50 subunit in STZ-treated mice [84]. Whether mirogabalin and cebranopadol share a similar activity, requires future research.

## Figures and Tables

**Figure 1 molecules-28-07862-f001:**
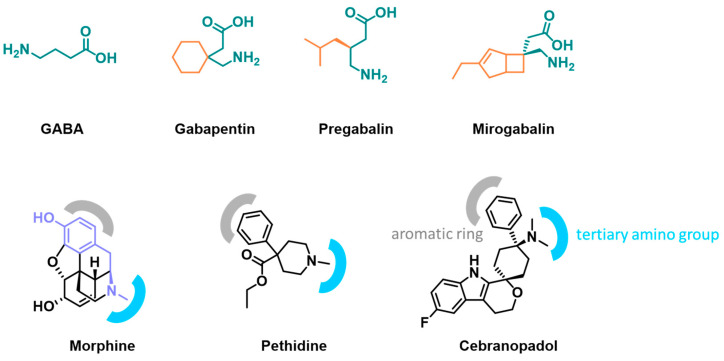
Structure of mirogabalin and cebranopadol and their analogs with marked key structural motives.

**Figure 2 molecules-28-07862-f002:**
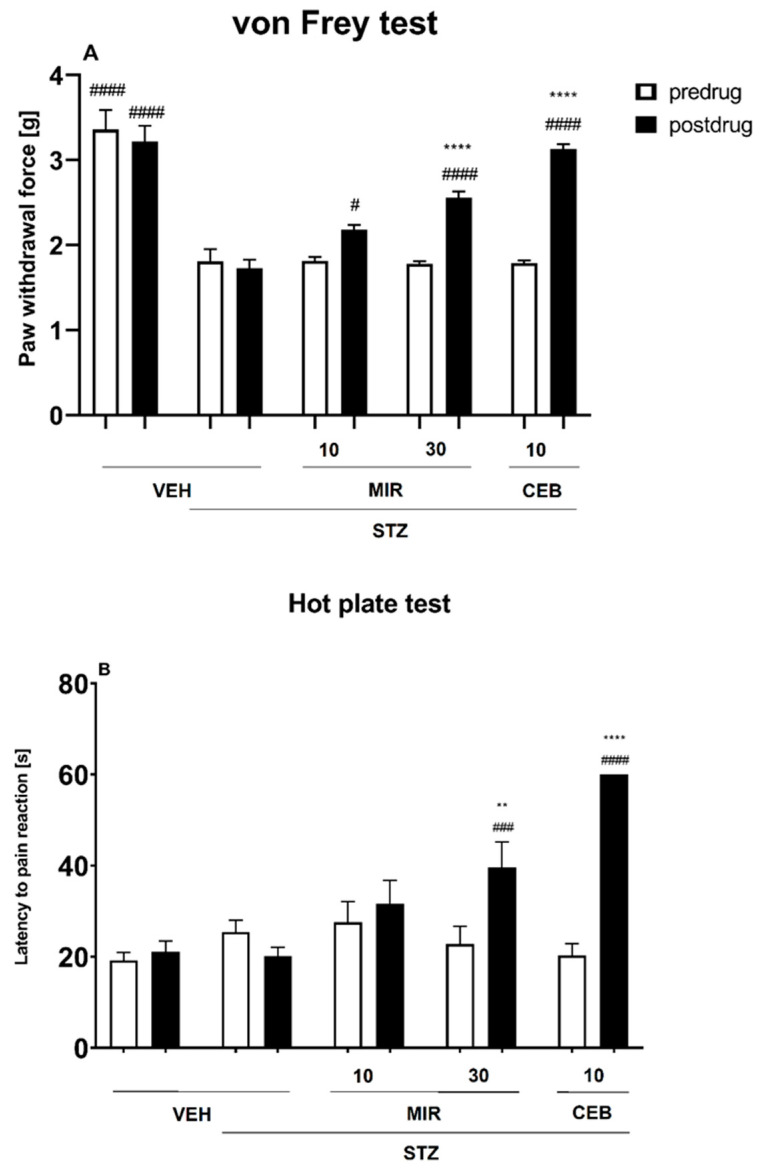
Effect of mirogabalin (MIR; doses: 10 and 30 mg/kg, i.p.) and cebranopadol (CEB; dose: 10 mg/kg, s.c.) on the mechanical (**A**) and heat (**B**) nociceptive threshold in the STZ-induced diabetic neuropathic pain model measured using the von Frey test (**A**) and the hot plate test (**B**), respectively. Results are shown as the mean paw withdrawal threshold [g] (±SEM) in response to mechanical stimulation (**A**), or the mean latency to pain reaction [s] (±SEM) in response to thermal stimulation (55–56 °C) (**B**) for *n* = 7–10. Statistical analysis: repeated measures analysis of variance (ANOVA), followed by Dunnett’s and Sidak’s post hoc comparison. Significance vs. STZ-treated control group (STZ + VEH): # *p* < 0.05, ### *p* < 0.001, #### *p* < 0.0001; significance vs. predrug paw withdrawal threshold, or predrug latency to pain reaction measured in the individual group: ** *p* < 0.01, **** *p* < 0.0001.

**Figure 3 molecules-28-07862-f003:**
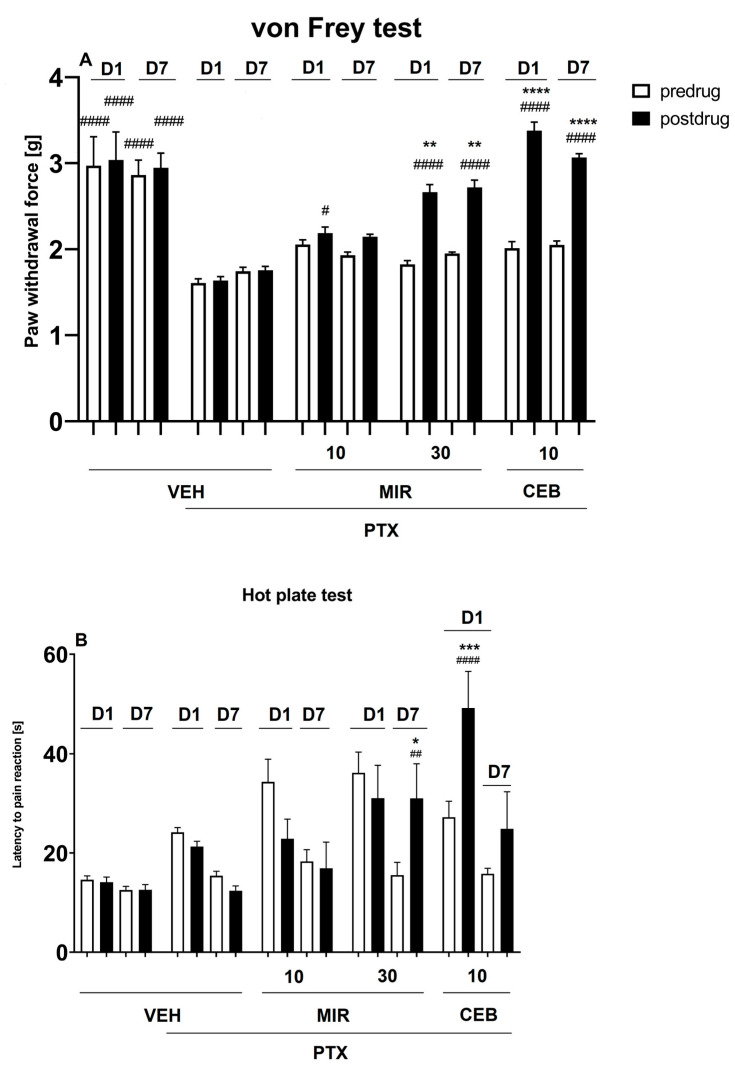
Effect mirogabalin (MIR; doses: 10 and 30 mg/kg, i.p.) and cebranopadol (CEB; dose: 10 mg/kg, s.c.) on the mechanical (**A**) and heat (**B**) nociceptive threshold measured in the paclitaxel (PTX)-induced neuropathic pain model using the von Frey test (**A**) and the hot plate test (**B**), respectively. Results are shown as the mean paw withdrawal threshold [g] (±SEM) in response to mechanical stimulation (**A**), or the mean latency to pain reaction [s] (±SEM) in response to thermal stimulation (55–56 °C) (**B**) measured for *n* = 7–10 on day 1 (D1) and day 7 (D7) of testing. Statistical analysis: repeated measures analysis of variance (ANOVA), followed by Dunnett’s and Sidak’s post hoc comparison. Significance vs. PTX-treated control group (PTX + VEH): # *p* < 0.05, ## *p* < 0.01, #### *p* < 0.0001; significance vs. predrug paw withdrawal threshold or predrug latency to pain reaction measured in the individual group: * *p* < 0.05, ** *p* < 0.01, *** *p* < 0.001, **** *p* < 0.0001.

**Figure 4 molecules-28-07862-f004:**
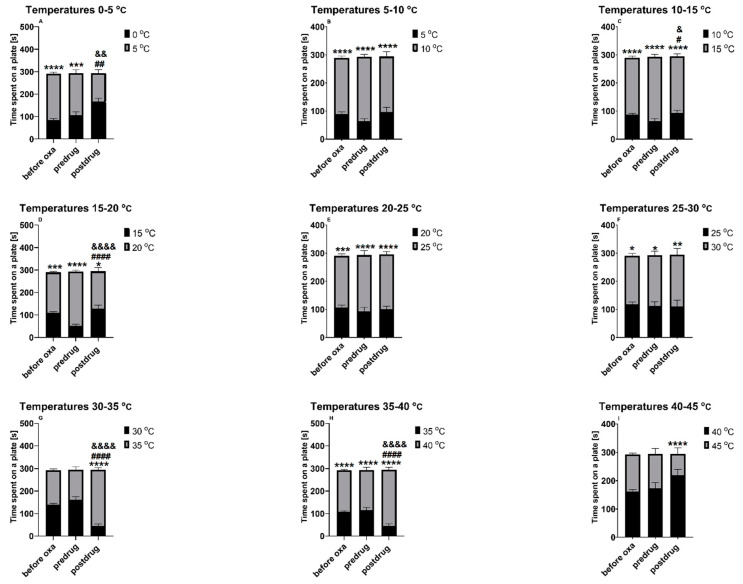
Effect of oxaliplatin (10 mg/kg) and mirogabalin (30 mg/kg) on the thermal place preference measured in mice. The two-plate thermal place preference test was conducted at various thermal ranges between 0 and 45 °C. Results are shown as time spent in a particular thermal zone (a plate) and measured at 3 distinct time points: before oxaliplatin injection (“before oxa”), 3 h after the injection of oxaliplatin (“predrug”) and 1 h after the intraperitoneal injection of mirogabalin (“postdrug”). Statistical analysis: repeated measures ANOVA and Sidak’s or Tukey’s multiple comparison. Significance vs. time [s] (±SEM) spent on the other plate at the same time point of testing: * *p* < 0.05, ** *p* < 0.01, *** *p* < 0.001, **** *p* < 0.0001; significance vs. time [s] (±SEM) spent on the plate set at a lower temperature and measured before mirogabalin administration (significance vs. predrug value): # *p* < 0.05, ## *p* < 0.01, #### *p* < 0.0001; significance vs. time [s] (±SEM) spent on the plate set at a higher temperature measured before mirogabalin administration (significance vs. predrug value): & *p* < 0.05, && *p* < 0.01, &&&& *p* < 0.0001. Oxa: oxaliplatin; *n* = 10.

**Figure 5 molecules-28-07862-f005:**
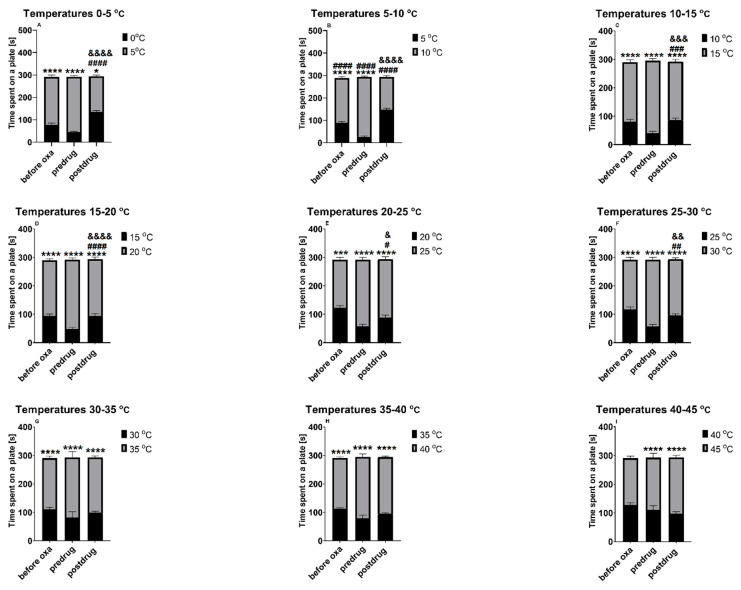
Effect of oxaliplatin (10 mg/kg) and cebranopadol (10 mg/kg) on the thermal place preference measured in mice. The two-plate thermal place preference test was conducted at various thermal ranges between 0 and 45 °C. Results are shown as time spent in a particular thermal zone (a plate) and measured at 3 distinct time points: before the oxaliplatin injection (“before oxa”), 3 h after the injection of oxaliplatin (“predrug”) and 90 min after the subcutaneous injection of cebranopadol (“postdrug”). Statistical analysis: repeated measures ANOVA and Sidak’s or Tukey’s multiple comparison. Significance vs. time [s] (±SEM) spent on the other plate at the same time point of testing: * *p* < 0.05, *** *p* < 0.001, **** *p* < 0.0001; significance vs. time [s] (±SEM) spent on the plate set at a lower temperature and measured before cebranopadol administration (significance vs. predrug value): # *p* < 0.05, ## *p* < 0.01, ### *p* < 0.001, #### *p* < 0.0001; significance vs. time [s] (±SEM) spent on the plate set at a higher temperature and measured before cebranopadol administration (significance vs. predrug value): & *p* < 0.05, && *p* < 0.01, &&& *p* < 0.001, &&&& *p* < 0.0001. Oxa: oxaliplatin; *n* = 10.

## Data Availability

The data presented in this study are available on request from the corresponding author.

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
