# Peer review of "Naturally Inspired Molecules for Neuropathic Pain Inhibition—Effect of Mirogabalin and Cebranopadol on Mechanical and Thermal Nociceptive Threshold in Mice"

_molecules, 2023, doi:10.3390/molecules28237862_

Round 1
Reviewer 1 Report
Comments and Suggestions for Authors
Salat et al. evaluated the effect of mirogabalin and cebranopadol on neuropathic pain. The results were mainly qualitative; however, they have merits. Further explanations and experimentation are required to validate their findings.
How many mice used in each group have to be mentioned (n=?)
The total number of mice used in the whole study must be mentioned
What food was provided to the mice has to be mentioned
Any anesthesia used in animals has to be mentioned
How much blood sample was collected has to be mentioned
AccuChek Active is not a recommended method to assess animal blood glucose. Authors may address this, or an alternative approach has to be adopted in the experiments.
If the chemical structures were copied from any source, that may be cited in the figure itself, or new images must be drawn using programs like Chemdraw.
At least, a well-established quantitative method (like COX-2 expression) in the different treatment groups has to be compared to the control mice to have a conclusive result on the effect of mirogabalin and cebranopadol in neuropathic pain management.
Author Response
Dear Reviewer,
Thank you for your comments regarding the manuscript. We have read them thoroughly and addressed them. In the revised manuscript, the changes and corrections made in the text are marked yellow.
Our response to your comments is as follows:
How many mice used in each group have to be mentioned (n=?)
The number of mice per group used in each neuropathic pain model is mentioned in Figs. 2-5 captions and in the section 4.1 (Animals and housing conditions): STZ and paclitaxel models: n=7-10; oxaliplatin model: n=10
The total number of mice used in the whole study must be mentioned
The total number of mice used was 295:
- in the STZ model 70 mice were involved at the beginning of the experiment but not all mice developed diabetes (only 70% were diabetic)
- in the paclitaxel model we used 45 mice
- in the oxaliplatin model 180 mice were used (90 for mirogabalin testing and 90 for cebranopadol testing)
What food was provided to the mice has to be mentioned
This is mentioned in the revised version of the manuscript, in the section 4.1 - The animals were kept in groups of 10 mice (…) and had free access to food (Rodentia Basic, Anima-Vivari, Poland) and water before experiments.
Any anesthesia used in animals has to be mentioned
This in vivo study, including the preparation of animals for the behavioral tests, was planned in such a way that the use of anesthesia was not necessary. This was because the use of both general and local anesthetics could significantly influence the results of tests assessing the pain threshold in animals (false positive results are likely). This research plan (without the use of anesthesia) was approved by the local ethical committee for animal experiments.
In addition to this, in order to cause the animals as little pain as possible and to comply with the ‘3R rule’ (Directive 2010/63/EU for protection of laboratory animals), the method of killing animals after behavioral tests was cervical dislocation, which causes immediate death of animals. Because of using a method of rapid euthanasia, there was no need to use anesthetics at this stage of experiment, as well.
How much blood sample was collected has to be mentioned
In the revised version of the manuscript this information is included in the section 4.3.1: Blood samples (5 µl/mouse) were obtained from the tail vein of the mice.
AccuChek Active is not a recommended method to assess animal blood glucose. Authors may address this, or an alternative approach has to be adopted in the experiments.
For measuring blood glucose levels we followed the method described by Tanabe and Murakami (Tanabe M, Murakami T, Ono H. Zonisamide suppresses pain symptoms of formalin-induced inflammatory and streptozotocin-induced diabetic neuropathy. J Pharmacol Sci. 2008 Jun;107(2):213-20. doi: 10.1254/jphs.08032fp.).
Although Accu-Chek Active system is generally dedicated for humans, it can also be easily used as a convenient and a rapid method for measuring glucose levels in mice. This device has a wide sensitivity range - up to 600 mg/dl, and values above are also saved – in this case the information "high" (blood glucose level) appears. In our research, at the stage of selecting animals for further behavioral tests, it was crucial to determine whether the glucose level was higher or lower than 300 mg/dl. For this purpose the Accu-Chek device is sensitive enough and we have been using it in our research since 2013, e.g.,:
- Sałat K, Gawlik K, Witalis J, Pawlica-Gosiewska D, Filipek B, Solnica B, Więckowski K, Malawska B. Evaluation of antinociceptive and antioxidant properties of 3-[4-(3-trifluoromethyl-phenyl)-piperazin-1-yl]-dihydrofuran-2-one in mice. Naunyn Schmiedebergs Arch Pharmacol. 2013 Jun;386(6):493-505. doi: 10.1007/s00210-013-0847-2.
- Sałat K, Kołaczkowski M, Furgała A, Rojek A, Śniecikowska J, Varney MA, Newman-Tancredi A. Antinociceptive, antiallodynic and antihyperalgesic effects of the 5-HT1A receptor selective agonist, NLX-112 in mouse models of pain. Neuropharmacology. 2017 Oct;125:181-188. doi: 10.1016/j.neuropharm.2017.07.022.
At present, it is also the only device we have got in our laboratory.
We would like to thank the Reviewer for this comment, we will consider purchasing a device dedicated for rodents in the nearest future, only if our financial resources allow for this purchase.
If the chemical structures were copied from any source, that may be cited in the figure itself, or new images must be drawn using programs like Chemdraw.
Chemical structures presented in the Fig. 1 are original and they were drawn by one of the co-authors (Dr. Paula Zaręba) with the use of ChemDraw 21.0.0 and modified using PowerPoint. This figure is not copied from any other sources.
At least, a well-established quantitative method (like COX-2 expression) in the different treatment groups has to be compared to the control mice to have a conclusive result on the effect of mirogabalin and cebranopadol in neuropathic pain management.
The main goal of our research was to conduct behavioral observations, because our group is focused on behavioral pharmacology. At this stage of the research, we wanted to check the effectiveness of both substances in 3 different models of neuropathic pain.
We did not plan to examine COX-2 expression because (in contrast to inflammatory pain conditions) it seems of little importance in neuropathic pain. We also did not examine the effects of mirogabalin and cebranopadol on other molecular targets crucial for the development of neuropathic pain, because such studies have been conducted previously and they have been published, e.g.,:
- Yamamura N, Mikkaichi T, Itokawa KI, Hoshi M, Damme K, Geigner S, Baumhauer C. Mirogabalin, a novel α2δ ligand, is not a substrate of LAT1, but of PEPT1, PEPT2, OAT1, OAT3, OCT2, MATE1 and MATE2-K. Xenobiotica. 2022 Sep-Nov;52(9-11):997-1009. doi: 10.1080/00498254.2022.2129517.
- Zajączkowska R, Pawlik K, Ciapała K, Piotrowska A, Ciechanowska A, Rojewska E, Kocot-Kępska M, Makuch W, Wordliczek J, Mika J. Mirogabalin Decreases Pain-like Behaviors by Inhibiting the Microglial/Macrophage Activation, p38MAPK Signaling, and Pronociceptive CCL2 and CCL5 Release in a Mouse Model of Neuropathic Pain. Pharmaceuticals (Basel). 2023 Jul 19;16(7):1023. doi: 10.3390/ph16071023.
- Komatsu S, Nakamura S, Nonaka T, Yamada T, Yamamoto T. Analgesic characteristics of a newly developed α2δ ligand, mirogabalin, on inflammatory pain. Mol Pain. 2021 Jan-Dec;17:17448069211052167. doi: 10.1177/17448069211052167.
- Kozai D, Numoto N, Nishikawa K, Kamegawa A, Kawasaki S, Hiroaki Y, Irie K, Oshima A, Hanzawa H, Shimada K, Kitano Y, Fujiyoshi Y. Recognition Mechanism of a Novel Gabapentinoid Drug, Mirogabalin, for Recombinant Human α2δ1, a Voltage-Gated Calcium Channel Subunit. J Mol Biol. 2023 May 15;435(10):168049. doi: 10.1016/j.jmb.2023.168049.
- Domon Y, Kobayashi N, Kubota K, Kitano Y, Ueki H, Shimojo Y, Ishikawa K, Ofune Y. The Novel Gabapentinoid Mirogabalin Prevents Upregulation of α2δ-1 Subunit of Voltage-Gated Calcium Channels in Spinal Dorsal Horn in a Rat Model of Spinal Nerve Ligation. Drug Res (Stuttg). 2023 Jan;73(1):54-60. doi: 10.1055/a-1941-8907.
- Ziemichod W, Kotlinska J, Gibula-Tarlowska E, Karkoszka N, Kedzierska E. Cebranopadol as a Novel Promising Agent for the Treatment of Pain. Molecules. 2022 Jun 21;27(13):3987. doi: 10.3390/molecules27133987.
For this reason, we didn't want to duplicate what had already been discovered by other researchers. We have mentioned this in the Conclusions section (lines 600 -606).
We do hope that the Reviewer will accept our explanations and will find this revised manuscript suitable for publication in the journal Molecules.
Kind regards
Kinga Sałat
Reviewer 2 Report
Comments and Suggestions for Authors
the research work is impressive for readers and for further
Line 17,18 and Line 58, 59. Both are same sentences, why the authors repeated
For what base the authors performed this work. Are they found any previous works related to this, if so, mention references with discussion at introduction
Abstract : Line 30-33 better these words to conclusion part. and write differently how your results useful to society and for further research
The drug names start with capital letter always
References seems cited old years, try to cite very recent works
The conclusion need amendments, the too much of discussion makes like results and discussion part. make it simple overall.
3. Discussion section: The authors discussed here their results along with some introduction like part with references. Am not sure if it is right or wrong to do like this.
My point is it is better to discuss your results discussion along with previous supporting results with references.
and compare how your results more positive over other with explanation and references.
Comments on the Quality of English Language
overall seems good.
Author Response
Dear Reviewer,
Thank you for your comments regarding the manuscript. We have read them thoroughly and addressed them. In the revised manuscript, the changes and corrections made in the text are marked yellow.
Our response to your comments and suggestions is as follows:
Line 17,18 and Line 58, 59. Both are same sentences, why the authors repeated?
Sentences in lines 17-18 are a part of the Abstract which needs to be able to stand alone, while sentences in lines 58-59 are a part of the main text. That was why they were similar. However, according to this Reviewer’s suggestion in this revised version of the manuscript we slightly changed these two sentences to avoid repetition. Please compare corrected sentences in lines 17-18 vs. 57-59.
For what base the authors performed this work. Are they found any previous works related to this, if so, mention references with discussion at introduction.
The justification for this research is mentioned both in the Introduction: e.g. :
- resistance of neuropathic pain to available analgesics, which necessitates the search for novel analgesic drugs (lines 44-47),
- promising results for mirogabalin and cebranopadol in some other pain models and their good safety profiles shown in previous studies (lines 87-95 and 136-141),
and in the Discussion section:
-efficacy and safety of these substances - the first two paragraphs: lines 309-336, 347 – 351,
- data about key role of VGCCs and opioid receptors in the development of neuropathic pain – lines 344-346, 404-410,
The rationale for this study is also mentioned in the Discussion in lines 315-322, 362-365, 374-376.
Abstract: Line 30-33 better these words to conclusion part and write differently how your results useful to society and for further research.
This is corrected in the revised manuscript (lines 30-33).
The drug names start with capital letter always.
We followed the style for INN drug names (can be reached at https://www.drugs.com/inn.html) and therefore for INN drug names we used lowercase letters for mirogabalin and cebranopadol.
References seems cited old years, try to cite very recent works.
We cited older papers mainly to refer to methodological details that were established many years ago (before the year 2000), and similarly these older papers were referred to as far as chemical details for opioid drugs developed before the year 2000 are concerned.
But to address this Reviewer’s comment, some newer papers have been added to the revised version of the manuscript, e.g.:
[12] Toll L, Cippitelli A, Ozawa A. The NOP Receptor System in Neurological and Psychiatric Disorders: Discrepancies, Peculiarities and Clinical Progress in Developing Targeted Therapies. CNS Drugs. 2021 Jun;35(6):591-607. doi: 10.1007/s40263-021-00821-0.
[21] Thouaye M, Yalcin I. Neuropathic pain: From actual pharmacological treatments to new therapeutic horizons. Pharmacol Ther. 2023 Nov;251:108546. doi: 10.1016/j.pharmthera.2023.108546.
[22] Yang S, Chang MC. Transcranial Direct Current Stimulation for the Management of Neuropathic Pain: A Narrative Review. Pain Physician. 2021 Sep;24(6):E771-E781.
[56] Xu Y, Jiang Z, Chen X. Mechanisms underlying paclitaxel-induced neuropathic pain: Channels, inflammation and immune regulations. Eur J Pharmacol. 2022 Oct 15;933:175288. doi: 10.1016/j.ejphar.2022.175288.
[71] Sałat K, Furgała A, Malikowska-Racia N. Searching for analgesic drug candidates alleviating oxaliplatin-induced cold hypersensitivity in mice. Chem Biol Drug Des. 2019 Jun;93(6):1061-1072. doi: 10.1111/cbdd.13507.
[72] Reis AS, Paltian JJ, Domingues WB, Novo DLR, Costa GP, Alves D, Campos VF, Mesko MF, Luchese C, Wilhelm EA. Advances in the Understanding of Oxaliplatin-Induced Peripheral Neuropathy in Mice: 7-Chloro-4-(Phenylselanyl) Quinoline as a Promising Therapeutic Agent. Mol Neurobiol. 2020 Dec;57(12):5219-5234. doi: 10.1007/s12035-020-02048-4.
[76] Yamamura, N.; Mikkaichi, T.; Itokawa, K.I.; Hoshi, M.; Damme, K.; Geigner, S.; Baumhauer, C. Mirogabalin, a novel α2δ ligand, is not a substrate of LAT1, but of PEPT1, PEPT2, OAT1, OAT3, OCT2, MATE1 and MATE2-K. Xenobiotica 2022, 52 (9-11), 997-1009. doi: 10.1080/00498254.2022.2129517.
[77] Komatsu, S.; Nakamura, S.; Nonaka, T.; Yamada, T.; Yamamoto, T. Analgesic characteristics of a newly developed α2δ ligand, mirogabalin, on inflammatory pain. Mol Pain 2021, 17, 17448069211052167. doi: 10.1177/17448069211052167.
[78] Domon, Y.; Kobayashi, N.; Kubota, K.; Kitano, Y.; Ueki, H.; Shimojo, Y.; Ishikawa, K.; Ofune, Y. The Novel Gabapentinoid Mirogabalin Prevents Upregulation of α2δ-1 Subunit of Voltage-Gated Calcium Channels in Spinal Dorsal Horn in a Rat Model of Spinal Nerve Ligation. Drug Res (Stuttg). 2023, 73 (1), 54-60. doi: 10.1055/a-1941-8907.
The conclusion need amendments, the too much of discussion makes like results and discussion part. make it simple overall.
This section (section 5 - Conclusions) has been corrected and modified in the revised version of the manuscript.
Discussion section: The authors discussed here their results along with some introduction like part with references. Am not sure if it is right or wrong to do like this.
This has been improved and shortened in the revised version of the manuscript – lines 315-328.
My point is it is better to discuss your results discussion along with previous supporting results with references and compare how your results more positive over other with explanation and references.
This has been improved in the revised version of the manuscript – lines: 318-322, 362-366, 374-376, 415-416.
We do hope that the Reviewer will accept our explanations and will find this revised manuscript suitable for publication in the journal Molecules.
Kind regards
Kinga Sałat
Round 2
Reviewer 1 Report
Comments and Suggestions for Authors
Thank you for the responses; however, I am not convinced with your statements and comments.
The number of mice n=7-10 animal/dose (as described in section 4.1) and in your response was found to be imprecise.
The total number of mice used in the study was found to be very high, even the authors did not undergo two or more independent experiments. I still have concerns on ethical committee approvals for the higher number of animals included in the study.
The exact statements on anesthesia and sacrifice of the mice was not included in the revised manuscript, as this may cause bias and genuineness of the ethic committee.
Checking the blood glucose level of rodents by AccuCheck is not a well-established method, and the authors collected 5μL of blood from the tail vein, as stated in their response. However, authors may have collected 50 μL blood from retro-orbital sinus and considered a standardized test would have more impact on their observations. In addition, quantitating the COX-2 expressions give a preliminary conclusive result on neuropathic pain in animals with the treatments, which several publications were available. The authors were not able to undergo those experiments, which may not open any conclusion.
Author Response
Our response to reviewer's comment is attached as a separate file.
